# Supplementary education can improve the rate of adequate bowel preparation in outpatients: A systematic review and meta-analysis based on randomized controlled trials

**Shicheng Peng[1][☯], Sixu Liu[1][☯], Jiaming Lei[1], Wensen Ren[1], Lijun Xiao[1], Xiaolan Liu[1], Muhan Lü[1‡]\*, Kai Zhou[2‡]\***

**1** Department of Gastroenterology, Affiliated Hospital of Southwest Medical University, Luzhou, China,
**2** Department of Emergency, Affiliated Hospital of Southwest Medical University, Luzhou, China

☯ These authors contributed equally to this work.
‡ These authors also contributed equally to this work
\* kaizhou7904@163.com (KZ); lvmuhan@swmu.edu.cn (ML)

**Data Availability Statement:** All relevant data are within the paper and its Supporting Information files.

## Abstract

### Background

Colonoscopy is widely used for the screening, diagnosis and treatment of intestinal diseases. Adequate bowel preparation is a prerequisite for high-quality colonoscopy. However, the rate of adequate bowel preparation in outpatients is low. Several studies on supplementary education methods have been conducted to improve the rate of adequate bowel preparation in outpatients. However, the controversial results presented encourage us to perform this meta-analysis.

### Method

According to the PRISMA statement (2020), the meta-analysis was registered on PROSPERO. We searched all studies up to August 28, 2021, in the three major electronic databases of PubMed, Web of Science and Cochrane Library. The primary outcome was adequate bowel preparation rate, and the secondary outcomes included bowel preparation quality score, polyp detection rate, adenoma detection rate, cecal intubation time, withdrawal time, nonattendance rate and willingness to repeat rate. If there was obvious heterogeneity, the funnel plot combined with Egger's test, meta-regression analysis, sensitivity analysis and subgroup analysis were used to detect the source of heterogeneity. RevMan 5.3 and Stata 17.0 software were used for statistical analysis.

### Results

A total of 2061 records were retrieved, and 21 full texts were ultimately included in the analysis. Our meta-analysis shows that supplementary education can increase the rate of adequate bowel preparation for outpatients (79.9% vs 72.9%, RR = 1.14, 95% CI: 1.08–1.20, $I^2$

**Funding:** Special support (cultivation) for young scientific and technological talents of Southwest Medical University (No. 00031718) and Talent development project of The Affiliated Hospital of Southwest Medical University (N0. 20061). There was no additional external funding received for this study. The funders had no role in study design, data collection and analysis, decision to publish, or preparation of the manuscript.

**Competing interests:** The authors have declared that no competing interests exist.

= 87%, $p$<0.00001). Supplementary education shortened the withdrawal time (MD: -0.80, 95% CI: -1.54 to -0.05, $p$ = 0.04) of outpatients, increased the Boston Bowel Preparation Scale (MD: 0.40, 95% CI: 0.36 to 0.44, $p$<0.00001), reduced the Ottawa Bowel Preparation Scale (MD: -1.26, 95% CI: -1.66 to -0.86, $p$<0.00001) and increased the willingness to repeat (91.9% vs 81.4%, RR:1.14, 95% CI: 1.04 to 1.25, $p$ = 0.004).

## Conclusion

Supplementary education for outpatients based on the standard of care can significantly improve the quality of bowel preparation.

## Introduction

Colonoscopy has been widely used in the inspection of polyps, adenomas, tumors, bleeding, inflammation, and stenosis [1]. Adequate visualization of the intestinal cavity is a prerequisite for high-quality colonoscopy [2]. Adequate bowel preparation can reduce the risk of prolonged procedure time, aborting procedures, repeated examinations, missed lesions, and delayed diagnosis, with avoidance of the waste of medical resources and medical insurance [3,4]. Inadequate bowel preparation increases the operating time and complication rates [5]. Even in recent years, the rate of inadequate bowel preparation is still as high as 35% [6]. Factors affecting the quality of intestinal preparation of patients include education level, sex, economic level, family relationship, tolerance of laxatives, professional level of instructors, patient comprehension and cooperative degree, previous abdominal or colonic surgery, diabetes mellitus obesity, chronic constipation, drugs (opioids, antidepressants) and neurologic diseases [7–11]. Usually, outpatients receive oral and written booklet instructions on bowel preparation when they make bowel preparation appointments. However, as early as 2001, research by Ness, R.M et al. found that such guidance often fails to achieve sufficient bowel preparation [8]. To increase the awareness of bowel preparation in outpatients and improve compliance, researchers have made extensive attempts. Examples included cartoon education booklets [12,13], educational videos [14–16], smartphone applications [17,18], telephone communication [7,19–21] and message reminders [22–24]. A recent meta-analysis showed that multimedia education can increase the rate of adequate bowel preparation and the detection rate of adenomas during colonoscopy [25]. A meta-analysis published in 2017 showed that these methods improved the quality of bowel preparation for colonoscopy [26]. However, several recent randomized controlled trials have found that these measures cannot improve the quality of intestinal preparation for outpatients [14,21–23]. To date, there is no meta-analysis on whether supplementary education can improve the rate of adequate bowel preparation for outpatients. Considering the contradictory results of multiple randomized controlled trials, we believe that it is necessary to complete such a systematic review and meta-analysis.

## Methods

This systematic review and meta-analysis was reported according to Preferred Reporting Items for Systematic Reviews and Meta-Analyses (PRISMA) 2020 [27] and registered on the International Prospective Register of Systematic Reviews (PROSPERO: CRD42021241308).

## Search strategy

With the help of librarians (BL) and statisticians (RC), the search terms were determined, and two researchers independently conducted comprehensive literature searches on the three major electronic databases (PubMed, Web of Science, and Cochrane Library). The search time started from the establishment of each database and ended on August 28, 2021. A comprehensive search was carried out using Medical Subject Heading+ Entrée terms, and the following search terms were used: "outpatient", "outpatients", "out-patients", "out patients", "out-patient", "bowel preparation" and "bowel cleansing". The search did not limit the language or type of research.

## Study screening

All search results were imported into EndNoteX9 (Thomson Corporation, Stanford, USA), and two researchers independently completed article screening according to the PRISMA 2020 flow diagram.

## Population

All adult outpatients who were scheduled for colonoscopy. Patients who had previously undergone surgical colorectal surgery or cognitive impairment were excluded.

Intervention: Considering the diverse methods of supplementary education, we did not restrict intervention measures when searching. Supplementary education included but was not limited to measures such as telephone calls, text messages, educational videos, smartphone applications, knowledge questionnaires and booklets that could increase the patient's understanding and compliance with bowel preparation. We did not restrict the laxatives used for bowel preparation.

## Comparison

Standard of care educational materials plus supplementary education with standard of care educational materials only. New intervention methods such as video, smartphone applications or network connections alone compared with standard of care educational materials were excluded.

## Outcome

Adequate bowel preparation rate based on the Boston Bowel Preparation Scale (BBPS), Ottawa Bowel Preparation Quality Scale (OBPQS), Aronchick Scale (ACS), Universal Preparation Assessment Scale (UPAS) and Harefield Cleansing Scale (HCS).

## Study

Prospective randomized controlled trial. Studies for which the full text was not available were excluded. For repeated research, the latest and most complete studies were selected.

## Outcomes

### Primary outcome

Adequate bowel preparation rate: the proportion of patients who considered adequate bowel preparation according to the scoring scale in each trial.

### Secondary outcomes

Bowel preparation quality score, polyp detection rate, adenoma detection rate, cecal intubation time, withdrawal time, nonattendance rate and willingness to repeat rate.

## Data extraction

The two researchers independently extracted the data included in the study into standardized forms. If there was a disagreement, it was discussed with the third researcher until an agreement was reached. The following data of the included studies were extracted: study first author, published year, country, research style, sample size, age, sex ratio(male/female), bowel preparation regimen, diet restriction, supplementary education method, quality evaluation scale, adequate bowel preparation rate (n/N, %), BBPS, OBPQS, polyp detection rate, adenoma detection rate, nonattendance rate and willingness to repeat rate. Taking into account the diversity of supplementary education, we try to classify the following according to the main characteristics: smartphone applications (whether it is social software such as WeChat's official account push or targeted development applications), video (regardless of whether the video acquisition form is offline or online), short messages (either serial or targeted), telephone calls (to communicate with patients via telephone voice) and booklets (booklets designed to increase patient understanding).

## Quality assessment

Two researchers independently conducted quality evaluations based on the Cochrane Collaboration's tool and the modified Jadad scale. Disagreements were resolved through discussion with a third researcher. The Cochran risk assessment tool makes high-risk, low-risk or unclear-risk judgments on random sequence generation, allocation concealment, blinding of participants and personnel, blinding of outcome assessment, incomplete outcome data, selective reporting and other sources of bias [28]. The modified Jadad scale scored from four aspects: randomization (0: Not randomized or inappropriate method of randomization, 1: The study was described as randomized, 2: The method of randomization was described and it was appropriate), concealment of allocation (0: Not describe the method of allocation concealment, 1: The study was described using the allocation concealment method, 2: The method of allocation concealment was described appropriately), double blinding (0: No blind or inappropriate method of blinding, 1: The study was described as double blind, 2. The method of double blinding was described, and it was appropriate), and withdrawals and dropouts (0: Not describe the follow-up, 1: A description of withdrawals and dropouts) [29]. Scores of 1–3 and 4–7 are considered low quality and high quality respectively.

## Statistical analysis

Since the bowel preparation laxatives, bowel quality evaluation scale and supplementary education are not completely consistent, and factors such as age, gender, and country may also have an impact, we used a random effects model for predictive analysis. Since the included studies were all randomized controlled trials, we used relative risk to conduct a meta-analysis of dichotomous data. Since each continuous variable meta-analysis is based on the same measurement unit, the weighted mean difference is used to measure the effect of each sample size and the 95% confidence interval. Considering the ceiling effect [30], the benefits of supplementary education were analyzed separately according to the adequate bowel preparation rate in the control group (<70%). A sensitivity analysis was carried out using a one-by-one elimination method to assess the robustness of the results. The $\chi^2$ test and $I^2$ statistics were used to assess heterogeneity. When $I^2 > 50\%$ and $P < 0.1$, it was considered that there was obvious heterogeneity [31]. When there was obvious heterogeneity and the number of studies was greater than or equal to twenty, meta-regression analysis was performed to explore the source of heterogeneity based on publication year, country, bowel preparation regimen, diet restriction, supplementary education method,

quality evaluation scale, and Jadad score. At the same time, a subgroup analysis was carried out based on the above factors. According to the publication year, sample size and Jadad score, a cumulative meta-analysis was carried out to explore the trend of research results. When the number of studies was greater than seven, the funnel plot and Egger's test were performed to evaluate publication bias. All statistical analysis were completed by Stata 17.0 MP-Parallel Edition (College Station, Texas, USA) and RevMan 5.3(London, United Kingdom).

## Results

Finally, 2062 records were retrieved, and EndnoteX9 excluded 361 duplicate records. After reading the title and abstract, 1613 records were excluded, 88 records were searched and the full text was carefully read. Finally, 21 articles (11028 patients) [7,12–24,32–38] were included in the analysis (Fig 1).

### Research basic characteristics

The characteristics of the 21 included studies are summarized in Table 1. All studies were randomized controlled trials, including four multicenter studies. Of the included studies, nine were from the United States, seven were from China, three were from Spain, and one was from Malaysia and Italy. Twenty of the included studies were published after 2010, and only one article was published in 2009. Only three studies had a sample size of less than 100, and seven studies had a sample size of more than 500. The bowel preparation regimens included 2 L polyethylene glycol (PEG)+ ascorbate solution, split dose 4 L PEG, 4 L PEG, split dose 3 L PEG and sodium phosphate. The supplementary education measures of the intervention group mainly included: smartphone application, short messages, telephone call, video and booklet. The bowel preparation quality evaluation scale includes: ACS, UPAS, BBPS and OBPQS. The Jadad scores of the included studies ranged from 1 to 6.

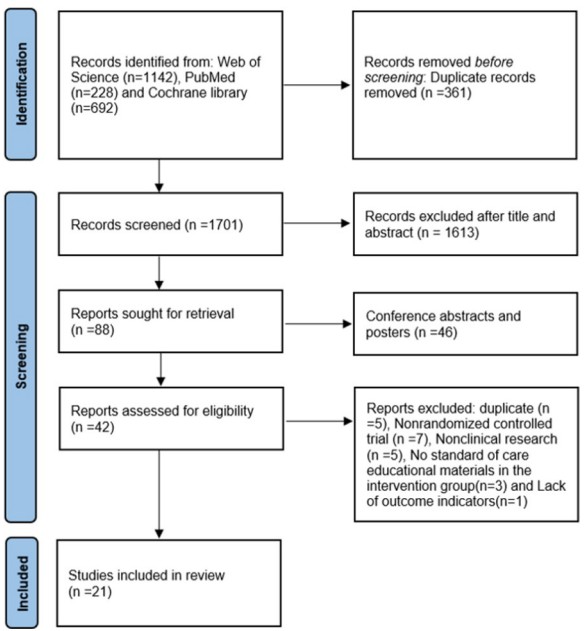

**Fig 1. Screening flowchart for the included studies.**

### Risk of bias in studies

The research quality evaluation based on the Jadad scale and the Cochran risk assessment manual is shown in Table 1 and Fig 2 and S1 Table and S1 Fig. According to the Jadad scale, fifteen of the studies included in the analysis were of high quality, and the remaining six were of low quality. Due to the particularity of the intervention measures of the study subjects, all studies can only achieve single-blind endoscope doctors.

## Primary outcome

### Adequate bowel preparation rates

As shown in Table 2, as the primary outcome, all studies (n = 21) reported adequate bowel preparation rates. Eleven of the studies [12,13,16,18,19,22–24,33,34,37] were in the low-ratio group(the adequate bowel preparation rate in the control group was less than 70%), and the remaining ten studies [7,14,15,17,20,21,32,35,36,38] were in the high-ratio group (the adequate bowel preparation rate in the control group reached or exceeded 70%) (Table 2). As shown in Fig 3, supplemental education increased adequate bowel readiness by 10.47% (60.53% to 71.9%, p< 0.00001) in the low-ratio group, but only 4.53% (82.67% to 87.20, p = 0.003) in the high-ratio group. Pooled analysis also showed that supplementary education significantly increased the rate of adequate bowel preparation (79.9% vs 72.9, RR = 1.14, 95% CI: 1.08–1.20, $I^2$ = 87%, p<0.00001) (Fig 4). Based on the $I^2$ and $p$ values, we believed that there was obvious heterogeneity in the research, and we explored the heterogeneity. As shown by the funnel plot (Fig 5) and Egger test's (p = 0.001), the study had obvious publication bias. Sensitivity analysis showed (S2 Fig) that when Sivakami Janahiraman's article [13] was removed, the research risk ratio changed the most, from 1.14 (95% CI: 1.08–1.20) to 1.10 (95% CI: 1.06–1.15), but it still failed to change the results. Then we improved the meta regression analysis based on the year of publication, country, bowel preparation regimen, diet restriction, supplementary education methods, quality evaluation scale, and Jadad score. As shown in Table 3, the bowel preparation regimen could explain 84.15% of the heterogeneity (p = 0.000). Next, we conducted cumulative meta-analysis based on the publication year, total sample size, and Jadad score. No obvious trend was found (S3–S5 Figs). Finally, we completed the subgroup analysis based on the above factors.

## Subgroup analysis

### Year

A considerable number of relevant studies had been completed in the past three years, and we completed the subgroup analysis within three years and three years ago. Ten studies [13,14,19–24,32,35] were published in the last three years. Compared with the control group, supplementary education significantly improved the rate of adequate bowel preparation for colonoscopy in outpatients (79.9% vs 72.2%, RR:1.13, 95% CI:1.05 to 1.22, $I^2$ = 88%, $p$ = 0.002) (S6A Fig). As shown in S6A Fig, eleven studies [7,12,15–18,33,34,36–38] were published before 2019, and supplementary education effectively increased the rate of adequate bowel preparation (81.5% vs 72.3%, RR:1.16, 95% CI:1.07 to 1.26, $I^2$ = 89%, $p$ = 0.0005).

### Country

The analysis of nine studies [12,14,16,23,24,33,35–37] completed in the USA shows that supplementary education can significantly improve the rate of adequate bowel preparation for outpatients (72.1% vs 67.6%, RR:1.09, 95% CI: 1.01 to 1.19, $I^2$ = 65%, $p$ = 0.03) (S6B Fig). A

**Table 1. Summary characteristics of studies included in the meta-analysis.**

| Study | Country | Research style | Research time | Sample size(n) | Age (years) | Sex (n, male/ female) | BPR | Diet restriction | SEM | QES | Jadad scale |
|---|---|---|---|---|---|---|---|---|---|---|---|
| Vicente Lorenzo-Zúñiga, 2015 [17] | Spain | Single center, RCT | Jan 2012 to Jun 2012 | 260 | ≥18 | 108/152 | 2L PEG + ascorbate solution | Low-fiber | Smartphone application | HCS | 1 |
| Thomas Y T Lam, 2020 [22] | China | Multicenter, RCT | Nov 2013 to Oct 2019 | 2225 | ≥18 | 1091/1134 | Split dose 4L PEG | Low-residue | Text messages | ACS | 1 |
| Alida Andrealli, 2018 [38] | Italy | Single center, RCT | Jan 2016 to Jun 2016 | 286 | 50–69 | 141/145 | 2L PEG + ascorbate solution | Low-residue | A brief counselling session | BBPS | 5 |
| Marco Antonio Alvarez-Gonzalez, 2020 [21] | Spain | Multicenter, RCT | Jan 2017 to Jun 2016 | 651 | 18–85 | 364/287 | Split dose 4L PEG | Low-fiber | Telephone call | BBPS | 5 |
| Ted B. Walker,2021 [14] | USA | Single center, RCT | - | 213 | ≥18 | 86/127 | - | - | Video | BBPS | 5 |
| Chunna Liu,2018 [15] | China | Single center, RCT | May 2016 to Oct 2017 | 476 | 18–80 | 301/175 | Split dose 4L PEG | Clear liquid | Video | OBPQS | 5 |
| Shashank Garg,2016 [37] | USA | Single center, RCT | Sep 2012 to Dec 2013 | 94 | ≥18 | 52/42 | 4L PEG | Clear liquid | Multimedia Education | ACS | 3 |
| Hong Shi,2019 [32] | China | Single center, RCT | Sep 2017 to Feb 2018 | 400 | 18–70 | 227/173 | Split dose 4L PEG | Low-residue | Smartphone application | BBPS | 5 |
| Nadim Mahmud, 2021 [23] | USA | Single center, RCT | Jan 2019 to Sep 2019 | 753 | 18–85 | 364/389 | Split dose 4L PEG | Clear liquid | Text messages | ACS | 5 |
| Xiaoyu Kang,2016 [18] | China | Multicenter, RCT | May 2014 to Nov 2014 | 770 | 18–80 | 393/377 | Split dose 4L PEG | - | Smartphone application | OBPQS | 5 |
| Agustín Seoane,2020 [20] | Spain | Single center, RCT | Nov 2017 to May 2018 | 1484 | ≥18 | 710/774 | - | Low-fiber | Telephone call | BBPS | 5 |
| Xiaodong Liu,2013 [7] | China | Single center, RCT | Feb 2012 to Jul 2012 | 605 | 18–75 | 307/298 | 4L PEG | Clear liquid | Telephone call | OBPQS | 5 |
| Brennan M.R. Spiegel,2011 [12] | USA | Single center, RCT | Sep 2009 to Dec 2009 | 436 | >18 | 423/13 | - | Clear liquid | Booklet | OBPQS | 6 |
| Chun-Jiu Hu,2021 [19] | China | Single center, RCT | Dec 2014 to Dec 2015 | 162 | ≥65 | 80/82 | 4L PEG | Semiliquid | Telephone call | OBPQS | 5 |
| Sivakami Janahiraman, 2020 [13] | Malaysia | Single center, RCT | - | 300 | ≥18 | 150/150 | Split dose 3L PEG | Low-residue | Booklet | BBPS | 5 |
| Audrey H. Calderwood, 2011 [36] | USA | Single center, RCT | Feb 2006 to Aug 2008 | 969 | ≥18 | 403/566 | - | - | Visual aid | BBPS | 5 |
| Sean C. Rice,2016 [16] | USA | Single center, RCT | Aug 2015 to Nov 2015 | 92 | ≥18 | 53/39 | Split dose 4L PEG | Clear liquid | Video | BBPS | 5 |
| Adeyinka O. Laiyemo, 2019 [35] | USA | Single center, RCT | Sep 2014 to Mar 2017 | 399 | ≥45 | 188/211 | Split dose 4L PEG | Clear liquid | Social contact | ACS | 5 |
| Feng-Chi Hsueh,2014 [34] | China | Single center, RCT | Jan 2011 to Apr 2011 | 218 | ≥20 | 116/102 | Sodium phosphate | Low-residue | video | ACS | 2 |
| Nadim Mahmud, 2019 [24] | USA | Single center, RCT | Apr 2018 | 71 | 18–75 | 37/34 | - | Clear liquid | Text messages | - | 3 |
| Chintan Modi,2009 [33] | USA | Multicenter, RCT | Jun 2007 to Jan 2008 | 164 | ≥40 | 65/99 | 4L PEG | Clear liquid | Test questionnaire | UPAS | 3 |

BPR: Bowel preparation regimen; SEM: Supplementary education method; QES: Quality Evaluation Scale; RCT: Randomized controlled trial; PEG: Polyethylene glycol; HCS: Harefield Cleansing Scale; ACS: Aronchick scale; UPAS: Universal Preparation Assessment Scale; BBPS: Boston Bowel Preparation Scale; OBPQS: Ottawa Bowel Preparation Quality Scale.

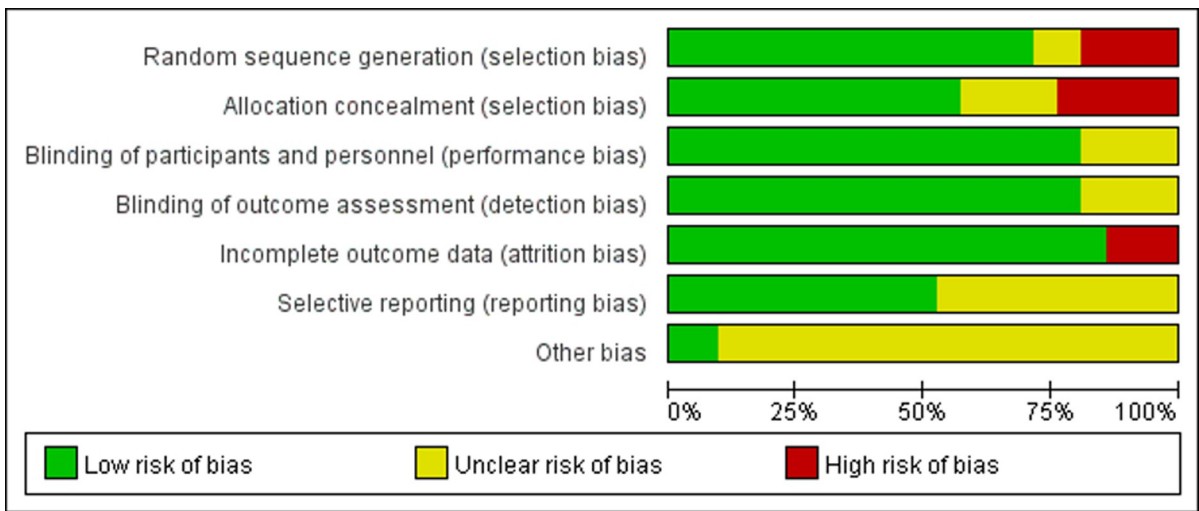

**Fig 2. Risk assessment graph based on Cochran's quality evaluation tool.**

subgroup analysis based on studies in China [7,15,18,19,22,32,34] (RR:1.19, 95% CI: 1.10 to 1.28, $I^2$ = 78%, $p$<0.00001) and Spain [17,20,21] (RR:1.03, 95%CI: 1.01 to 1.06, $I^2$ = 0%, $p$ = 0.007) also showed that supplementary education can significantly increase the rate of adequate bowel preparation (China: 78.7% vs 68.7% and Spain: 88.8% vs 85.8%) (S6B Fig).

## Bowel preparation regimen

The results of subgroup analysis based on different bowel preparation regimens showed that supplementary education in the 2 L PEG+ ascorbate solution (RR:1.02, 95% CI: 0.97 to 1.07, $I^2$ = 60%, $p$ = 0.44) group [17,38] could not improve the adequate bowel preparation rate of outpatients (S6C Fig). However, in the split-dose 4 L PEG [15,16,18,21–23,32,35] (75.0% vs 68.7%, RR:1.10, 95% CI: 1.05 to 1.15, $I^2$ = 43%, $p$<0.00001) and 4 L PEG [7,19,33,37] (78.1% vs 64.6%, RR:1.23, 95% CI: 1.10 to 1.38, $I^2$ = 31%, $p$ = 0.0004) groups, supplementary education improved the rate of adequate bowel preparation (S6C Fig).

## Diet restriction

Seventeen studies [7,12,13,15–17,20–24,32–35,37,38] reported on diet restriction in bowel preparation (Table 2). Subgroup analysis based on diet restriction types showed that supplementary education in the clear liquid diet group (73.3% vs 62.8%, RR:1.17, 95% CI: 1.09 to 1.27, $I^2$ = 57%, $p$<0.0001) and low-fiber/residue diet group (82.8% vs 75.6%, RR:1.14, 95% CI: 1.06 to 1.24, $I^2$ = 93%, $p$ = 0.001) increased the rate of adequate bowel preparation (S6D Fig).

## Supplementary education methods

Subgroup analysis based on video [14–16,34] (RR:1.21, 95% CI: 0.98 to 1.48, $I^2$ = 90%, $p$ = 0.07), short message [22–24] (RR:1.05, 95% CI: 0.97 to 1.13, p = 0.24) and smartphone application [17,18,32] (RR:1.10, 95% CI: 0.99 to 1.22, $I^2$ = 90%, $p$ = 0.09) as a supplementary educational method showed that there was no significant difference in the adequate bowel preparation rate between the two groups of outpatients (S6E Fig). The results of four telephone call [7,19–21] (86.0% vs 79.6%, RR:1.12, 95% CI: 1.01 to 1.25, $I^2$ = 84%, $p$ = 0.03) and two booklet [12,13] (80.5% vs 51.3%, RR:1.60, 95% CI: 1.15 to 2.23, $I^2$ = 89%, $p$ = 0.006)

**Table 2. Summary outcome indicators of studies included in the meta-analysis.**

| Study | ABP (n/N, %) | | BBPS (mean ± SD) | | OBPQS (mean± SD) | | CIT (min, mean± SD) | | WDT (min, mean± SD) | | PDR (n/N, %) | | ADR (n/N, %) | | NAR (n/N, %) | | WTRR (n/N, %) | |
|---|---|---|---|---|---|---|---|---|---|---|---|---|---|---|---|---|---|---|
| | int | con | int | con | int | con | int | con | int | con | int | con | int | con | int | con | int | con |
| Vicente Lorenzo-Zúñiga,2015 [17] | 108/108, 100 | 146/152, 96.1 | - | - | - | - | - | - | - | - | - | - | - | - | - | - | 96/108, 88.9 | 116/152, 76.3 |
| Thomas Y T Lam, 2020 [22] | 687/983, 69.9 | 665/1010, 65.9 | - | - | - | - | - | - | - | - | - | - | - | - | 67/1050, 6.4 | 100/1110, 9.0 | - | - |
| Alida Andrealli,2018 [38] | 136/143, 95.1 | 137/143, 95.8 | 8.1± 1.2 | 7.8 ±1.4 | - | - | - | - | - | - | 77/ 143, 53.8 | 79/ 143, 55.2 | 52/ 143, 36.4 | 57/ 143, 39.9 | - | - | - | - |
| Marco Antonio Alvarez-Gonzalez,2020 [21] | 249/322, 77.3 | 237/329, 72.0 | - | - | - | - | - | - | - | - | - | - | 130/ 303, 42.9 | 117/ 302, 38.7 | 19/322, 5.9 | 27/329, 8.2 | - | - |
| Ted B. Walker,2021 [14] | 103/111, 92.8 | 94/102, 92.2 | 8.0 ±0.1 | 7.6 ±0.2 | - | - | - | - | - | - | 62/ 111, 55.9 | 65/ 102, 63.7 | 47/ 111, 42.3 | 49/ 102, 48.0 | 16/138, 11.6 | 20/131, 15.3 | - | - |
| Chunna Liu,2018 [15] | 215/239, 90.0 | 178/237, 75.1 | - | - | 3.05 ±1.3 | 4.18 ±1.4 | 5.1 ±4.8 | 6.0 ±4.2 | 6.8 ±2.5 | 7.0 ±3.2 | 32/ 239, 13.4 | 31/ 237, 13.1 | - | - | 23/262, 8.8 | 25/262, 9.5 | - | - |
| Shashank Garg,2016 [37] | 34/48, 70.8 | 22/46, 47.8 | - | - | - | - | - | - | - | - | 23/48, 47.9 | 16/46, 34.8 | 16/48, 33.3 | 9/46, 19.6 | 7/55, 12.7 | 2/48, 4.2 | - | - |
| Hong Shi,2019 [32] | 188/200, 94.0 | 174/200, 87.0 | - | - | - | - | - | - | - | - | - | - | - | - | - | - | - | - |
| Nadim Mahmud,2021 [23] | 195/367, 53.1 | 210/386, 54.4 | - | - | - | - | - | - | - | - | - | - | - | - | 49/367, 13.4 | 50/386, 13.0 | - | - |
| Xiaoyu Kang,2016 [18] | 318/387, 82.2 | 266/383, 69.5 | - | - | 3.6 ±1.7 | 4.5 ±1.8 | 7.2 ±4.6 | 9.1 ±4.8 | 7.2 ±2.2 | 7.4 ±2.1 | - | - | 72/ 387, 18.6 | 46/ 383, 12.0 | - | - | 324/353, 91.8 | 285/352, 81.0 |
| Agustín Seoane,2020 [20] | 622/673, 92.4 | 567/627, 90.4 | - | - | - | - | - | - | - | - | - | - | - | - | 62/738, 8.4 | 107/746, 14.3 | - | - |
| Xiaodong Liu,2013 [7] | 249/305, 81.6 | 211/300, 70.3 | - | - | 3.0 ±2.3 | 4.9 ±3.2 | 7.7 ±5.1 | 7.6 ±4.3 | 6.2 ±2.3 | 7.8 ±2.8 | 116/ 305, 38.0 | 74/ 300, 24.7 | - | - | 27/305, 8.9 | 21/300, 7.0 | 245/276, 88.8 | 236/273, 86.4 |
| Brennan M.R. Spiegel,2011 [12] | 147/216, 68.1 | 101/220, 45.9 | - | - | 4.4 ±2.3 | 5.1 ±2.9 | - | - | - | - | - | - | - | - | 33/216, 15.3 | 31/220, 14.1 | - | - |
| Chun-Jiu Hu,2021 [19] | 69/83, 83.1 | 47/79, 59.5 | - | - | 3.2 ±2.1 | 5.2 ±2.8 | 5.0 ±3.2 | 5.4 ±3.7 | 8.0 ±1.2 | 9.2 ±2.2 | 46/83, 55.4 | 32/79, 40.5 | - | - | - | - | - | - |
| Sivakami Janahiraman,2020 [13] | 147/149, 98.7 | 79/151, 52.3 | - | - | - | - | - | - | - | - | 64/ 149, 43.0 | 19/ 151, 12.6 | - | - | - | - | 149/149, 100 | 118/151, 78.1 |
| Audrey H. Calderwood,2011 [36] | 375/477, 78.6 | 393/492, 79.9 | - | - | - | - | - | - | - | - | 182/ 477, 38.2 | 189/ 492, 38.4 | - | - | - | - | - | - |
| Sean C. Rice,2016 [16] | 31/42, 73.8 | 34/50, 68.0 | - | - | - | - | - | - | - | - | - | - | - | - | - | - | - | - |
| Adeyinka O. Laiyemo,2019 [35] | 139/156, 89.1 | 123/152, 80.9 | - | - | - | - | - | - | - | - | - | - | - | - | 45/201, 22.4 | 46/198, 23.2 | - | - |
| Feng-Chi Hsueh,2014 [34] | 84/104, 80.8 | 55/114, 48.2 | - | - | - | - | - | - | - | - | - | - | - | - | - | - | - | - |
| Nadim Mahmud,2019 [24] | 16/21, 76.2 | 30/50, 60.0 | - | - | - | - | - | - | - | - | - | - | - | - | 0/21, 0 | 5/50, 10.0 | - | - |
| Chintan Modi,2009 [33] | 58/84, 69.0 | 46/80, 57.5 | - | - | - | - | - | - | - | - | - | - | - | - | - | - | - | - |

ABP: Adequate bowel preparation; BBPS: Boston Bowel Preparation Scale; OBPQS: Ottawa Bowel Preparation Quality Scale; CIT: Cecal intubation time; WDT: Withdrawal time; PDR: Polyp detection rate ADR: Adenoma detection rate; NAR: Nonattendance rate; WTRR: Willingness to repeat rate; int: Intervention group; con: Control group.

studies showed that supplementary education can significantly improve the rate of adequate bowel preparation for outpatients (S6E Fig).

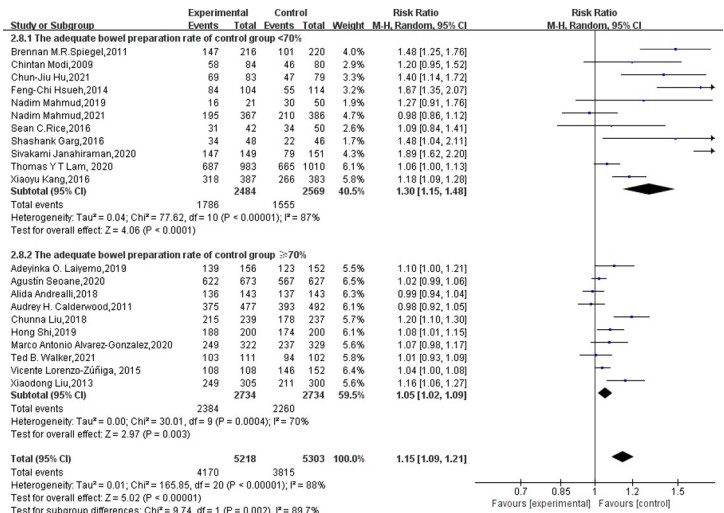

**Fig 3. Forest plots analyzed based on whether the adequate bowel preparation rate in the control group was below 70%.**

## Quality evaluation scale

As shown in S6F Fig, whether it is based on ACS [22,23,34,35,37] (68.7% vs 63.0%, RR:1.17, 95% CI: 1.02 to 1.34, $I^2 = 82\%$, $p = 0.02$), BBPS [13,14,16,20,21,32,36,38] (87.4% vs 82.0%, RR:1.09, 95% CI: 1.01 to 1.18, $I^2 = 91\%$, $p = 0.03$) or OBPQS [7,12,15,18,19] (81.1% vs 65.9%, RR:1.19, 95% CI: 1.14 to 1.25, $I^2 = 55\%$, $p<0.00001$), supplementary education could significantly improve the rate of adequate bowel preparation for outpatients.

## Jadad score

We conducted subgroup analysis according to the quality of the study based on the results of the Jadad score. The results of six low-quality (Jadad 1–3) studies [17,22,24,33,34,37] showed

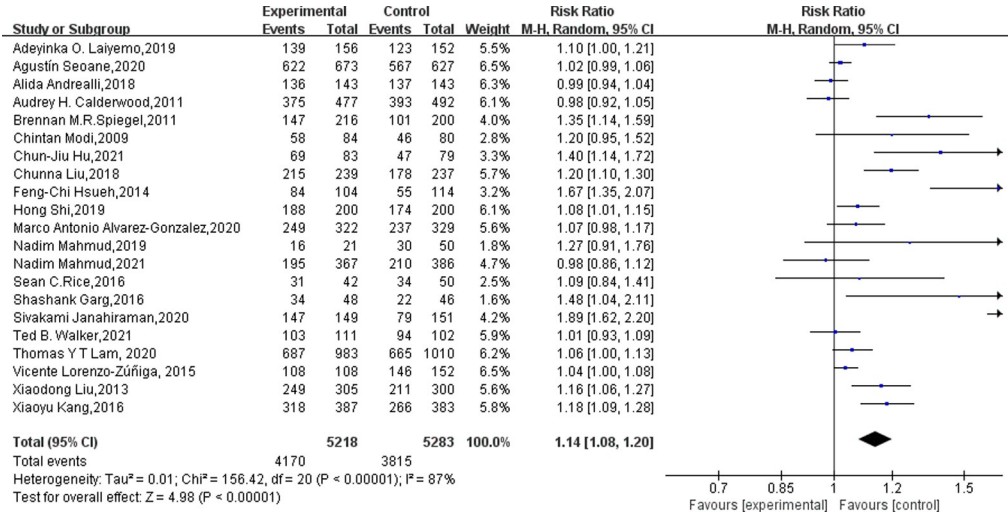

**Fig 4. Forest plot comparing the effects of supplementary education based on traditional education and traditional education alone on the adequate bowel preparation rate.**

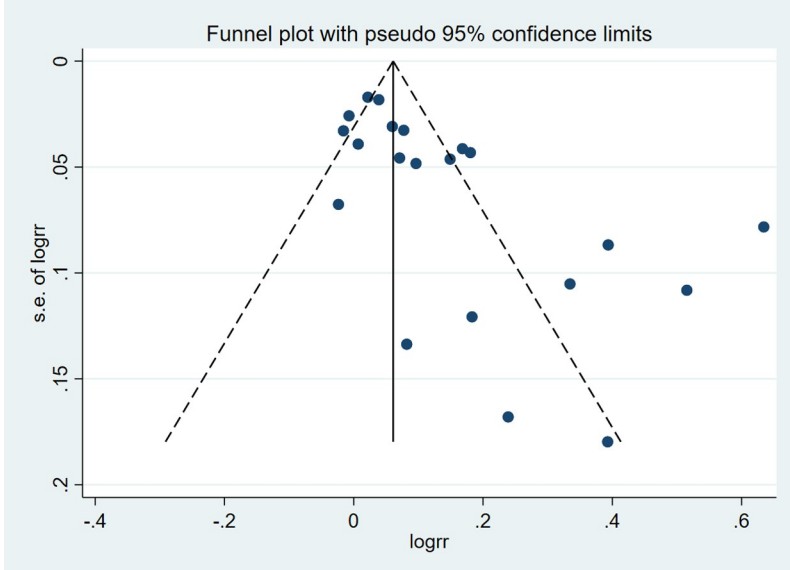

**Fig 5. Funnel plot comparing the effects of supplementary education based on traditional education and traditional education alone on the adequate bowel preparation rate.**

that supplementary education can increase the rate of adequate bowel preparation for outpatients (73.2% vs 66.4%, RR:1.21, 95% CI: 1.06 to 1.37, $I^2$ = 86%, $p$ = 0.004) (S6G Fig). The results of fifteen high-quality (Jadad 4–7) studies [7,12–16,18–21,23,32,35,36,38] also showed that supplementary education can improve patients' adequate bowel preparation rate (82.2% vs 74.4%, RR:1.13, 95% CI: 1.06 to 1.20, $I^2$ = 89%, $p$ = 0.0001) (S6G Fig).

## Secondary outcomes

### Bowel preparation quality score

Only two studies [14,38] reported the mean and standard deviation of BBPS scores in outpatients (Table 2). As shown in Fig 6A, supplementary education increased the colonoscopy BBPS score of outpatients (MD: 0.40, 95% CI: 0.36 to 0.44, $I^2$ = 0%, $p$<0.00001). A meta-analysis based on five reported OBPQS studies [7,12,15,18,19] showed that supplementary education can reduce colonoscopy OBPQS (Fig 6B) (MD: -1.26, 95% CI: -1.66 to -0.86, $I^2$ = 82%, $p$<0.00001).

### Cecal intubation time

As shown in Table 2, four studies [7,15,18,19] from China reported the average and standard deviation of the cecal intubation time. A meta-analysis based on the four studies showed that

**Table 3. Meta-regression analysis summary.**

| Covariates | Tau2 | I-squared res (%) | Adj R-squared (%) | P>\|t\| | 95% Conf. Interval |
|---|---|---|---|---|---|
| Year | 0.02662 | 86.34 | -4.23 | 0.531 | 0.9695975, 1.01659 |
| Country | 0.02693 | 86.41 | -5.43 | 0524 | 0.9445658, 1.114495 |
| Bowel preparation regimen | 0.004234 | 62.35 | 84.15 | 0.000 | 1.099721, 1.222451 |
| Supplementary education Method | 0.02603 | 86.40 | -1.92 | 0.294 | 0.984949, 1.048576 |
| Quality evaluation scale | 0.02678 | 86.26 | -4.86 | 0.612 | 0.9517594, 1.085227 |
| Jadad score | 0.02751 | 86.37 | -7.70 | 0.911 | 0.9401236, 1.056994 |
| Diet restriction | 0.02956 | 84.37 | -3.70 | 0.541 | 0.8919588, 1.233576 |

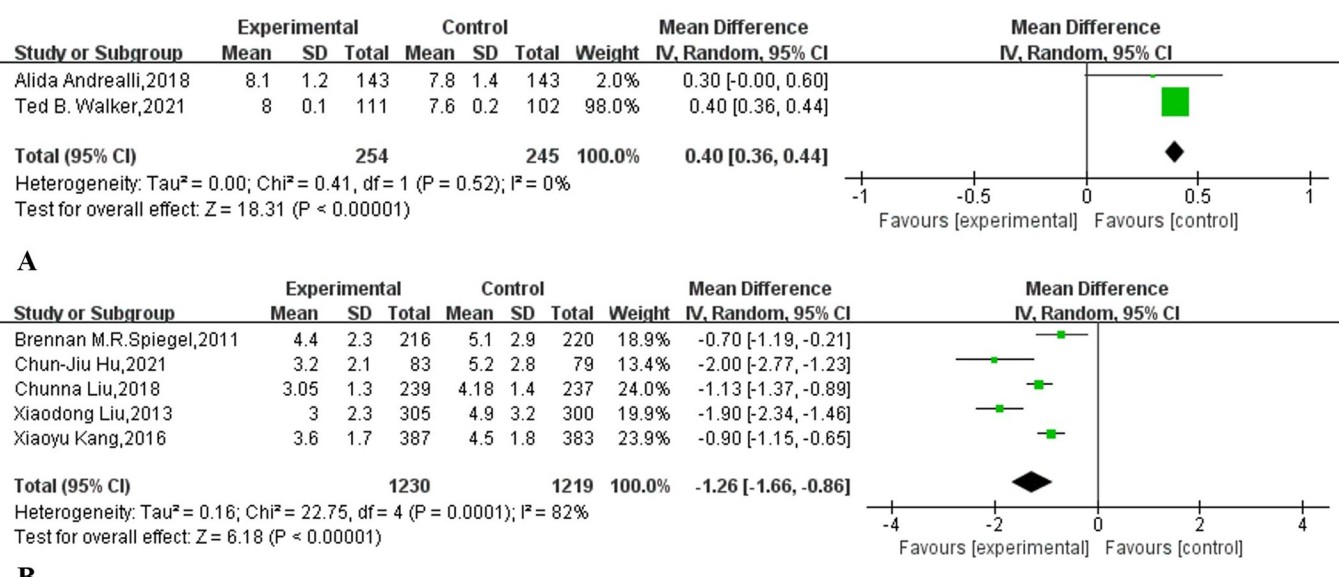

**Fig 6.** Forest plot comparing (A) the Boston Bowel Preparation Scale (BBPS) and Ottawa Bowel Preparation Quality Scale (OBPQS) supplementary education combined with traditional education and traditional education alone.

supplementary education did not significantly shorten the cecal intubation time (MD: -0.80, 95% CI: -1.74 to 0.14, $I^2$ = 82%, $p$ = 0.10) (Fig 7A).

## Withdrawal time

A meta-analysis based on four reported withdrawal time studies [7,15,18,19] showed that supplementary education can effectively shorten the withdrawal time (MD: -0.80, 95% CI: -1.54 to -0.05, $I^2$ = 92%, $p$ = 0.04) (Fig 7B).

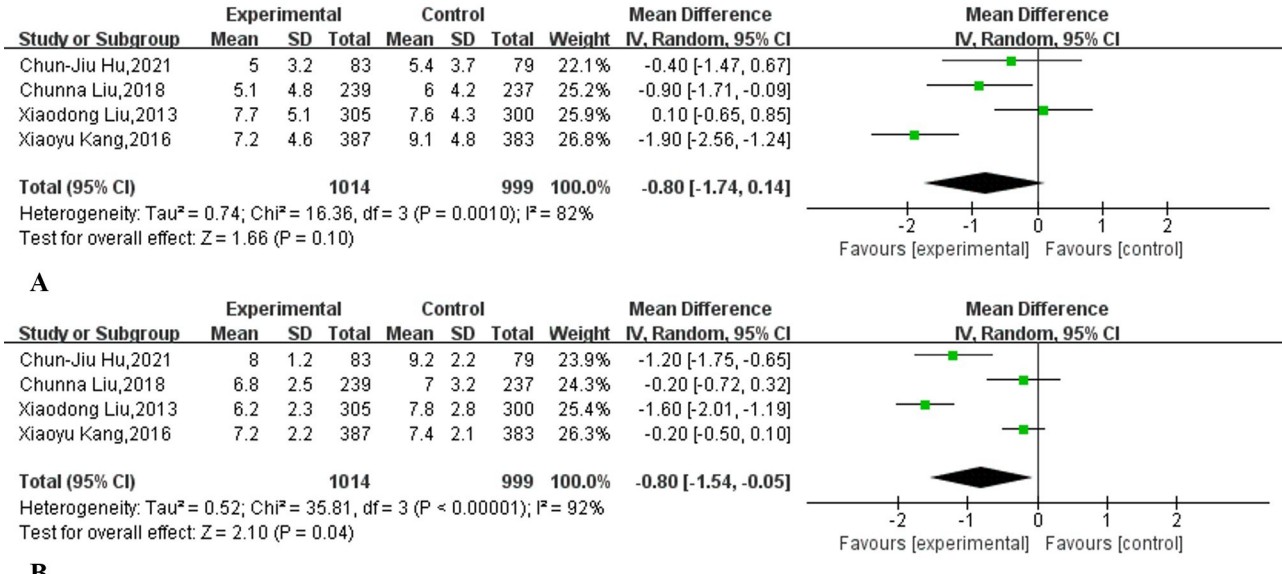

**Fig 7.** Forest plot comparing (A) cecal intubation time (CIT) and (B) withdrawal time (WT) supplementary education combined with traditional education and traditional education alone.

## Polyp detection rate

As shown in Table 2, eight studies [7,13–15,19,36–38] reported the detection rate of polyps. A meta-analysis based on these eight studies showed that supplementary education was not statistically significant in improving the detection rate of polyps under colonoscopy (RR:1.26, 95% CI: 0.99 to 1.60, $I^2$ = 83%, $p$ = 0.06) (Fig 8A). The funnel plot (S7 Fig) and Egger's test (p = 0.180) based on these eight studies did not find significant publication bias.

## Adenoma detection rate

Five studies [14,18,21,37,38] reported on the detection rate of adenomas (Table 2), and a meta-analysis based on these five studies showed that supplementary education did not improve the detection rate of adenomas under colonoscopy (RR:1.11, 95% CI: 0.90 to 1.38, $I^2$ = 56%, $p$ = 0.33) (Fig 8B).

## Nonattendance rate

As shown in Table 2, eleven studies [7,12,14,15,20–24,35,37] reported the nonattendance rate of colonoscopy. A meta-analysis based on these eleven studies showed that supplementary education cannot significantly reduce the nonattendance rate of colonoscopy in outpatients (RR:0.86, 95% CI: 0.71 to 1.03, $I^2$ = 38%, $p$ = 0.10) (Fig 9A). The funnel plot (S8 Fig) and Egger's test (p = 0.324) based on these eleven studies did not find significant publication bias.

## Willingness to repeat rate

Four studies [7,13,17,18] reported the willingness to repeat rate (Table 2). A meta-analysis based on these four studies showed that supplementary education can significantly increase

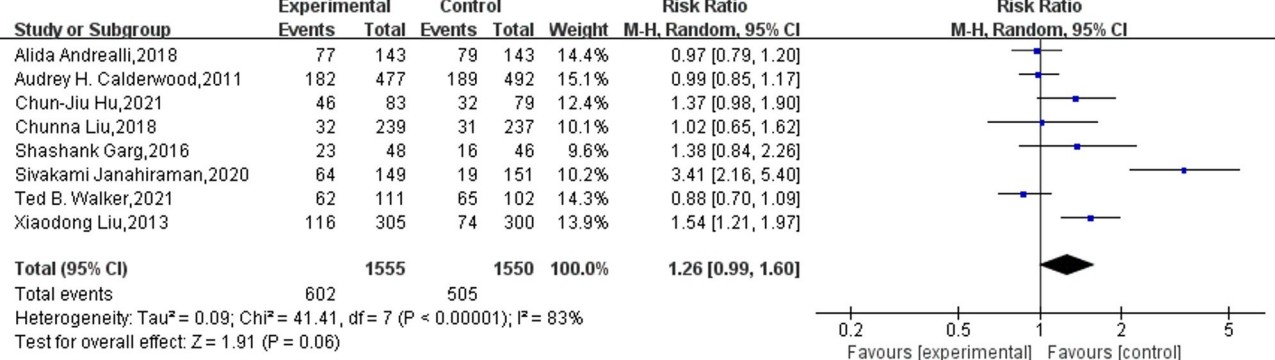

**Fig 8.** Forest plot comparing (A) polyp detection rate (PDR) and (B) adenoma detection rate (ADR) supplementary education combined with traditional education and traditional education alone.

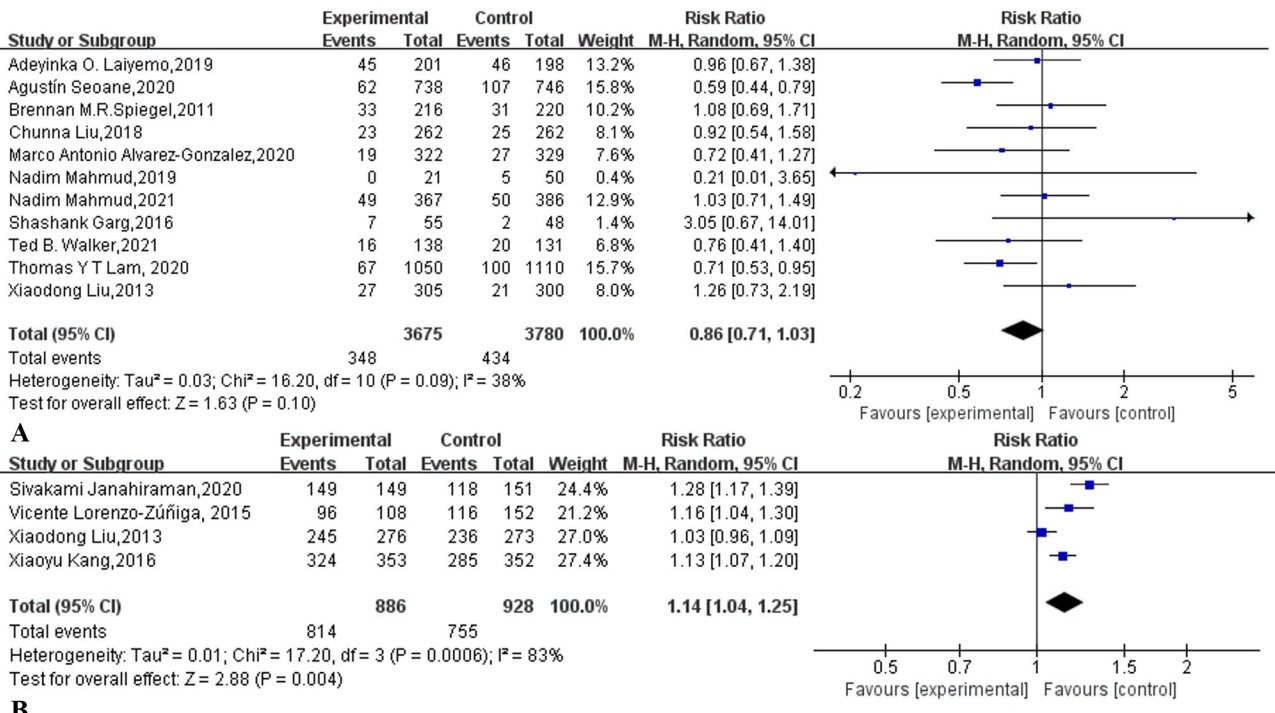

**Fig 9.** Forest plot comparing (A) nonattendance rate (NAR) and (B) willingness to repeat rate (WTRR) supplementary education combined with traditional education and traditional education alone.

the willingness to repeat rate of outpatients (91.9% vs 81.4%, RR:1.14, 95% CI: 1.04 to 1.25, $I^2 =$ 83%, $p$ = 0.004) (Fig 9B).

## Discussion

Adequate bowel preparation is not only a prerequisite for high-quality colonoscopy but also an important guarantee for colonoscopy safety [39]. A ceiling effect in bowel preparation is present [30]. In an unselected population, it is very difficult to improve adequate bowel preparation higher than a given rate (90%-95%). That is, in general, low rates of bowel preparation may be improved better than high rates. Our analysis confirmed this claim (Fig 3). Therefore, it may be more meaningful to implement supplementary education in areas or populations with low rates of adequate bowel preparation. Our pooled analysis shows that supplementary education based on traditional nursing education can significantly improve the rate of adequate bowel preparation (79.9% vs 72.9, $p$<0.00001) for outpatients. A recent meta-analysis showed that reinforced education based on standard education improves the quality of bowel preparation for colonoscopy [40]. Unlike this study, we only explored the effect of supplemental education on the quality of bowel preparation in outpatients. In addition, we included a larger number of studies and more cases. Considering that bowel preparation is not limited to colonoscopy and reinforcement methods are not necessarily named "education", we did not use "colonoscopy" and "education" as search terms to avoid omission. This is also consistent with several related meta-analyses published previously [25,26,41,42]. Due to the obvious heterogeneity ($I^2$ = 87%, $p$<0.00001), we analyzed the source of the heterogeneity. First, we completed meta-regression analysis for the publication year, country, bowel preparation regimen, diet restriction, supplementary education method, quality evaluation scale, and Jadad score. As shown in Table 3, the bowel preparation regimen accounted for most of the heterogeneity (Adj R-squared 84.15%,

p = 0.000). It is especially noteworthy that the value of Tau2 is also very low (0.004234), indicating a high level of confidence. Next, we also conducted a cumulative meta-analysis for the publication year, total sample size, and Jadad score. As shown in S3–S5 Figs, the above factors had no obvious trend in the impact of the research results. Then, the sensitivity analysis showed that no studies could significantly change the meta-analysis results (S2 Fig). Finally, we conducted subgroup analysis based on the characteristics of different factors, such as publication year, country, bowel preparation regimen, diet restriction, supplementary education method, quality evaluation scale, and Jadad score. As shown in S6 Fig, country, bowel preparation regimen, quality evaluation scale and supplementary education method can explain some of the sources of heterogeneity. Based on the results of the funnel plot combined Egger's test, meta-regression analysis, sensitivity analysis, cumulative meta-analysis and subgroup analysis, we believe that research heterogeneity is caused by publication bias and different bowel preparation regimens. Reasons for publication bias include the following: studies with positive or statistically significant results are more likely to be published than those with negative or insignificant results [43–45], authors are more likely to publish studies with positive results in English-language journals [44,46,47], and authors are selective about the results reported by the protocol hide [48–51]. Research suggests that conducting a prospective meta-analysis may address these concerns [52]. It is worth noting that under certain circumstances, supplementary education is not statistically significant in improving the rate of adequate bowel preparation for outpatients. For example, 2 L PEG+ ascorbate solution was used as a bowel preparation regimen, and videos, short messages and smartphone applications were used as supplementary educational methods. Of course, whether supplementary education is meaningless in improving the rate of adequate bowel preparation in outpatients under these circumstances remains to be further studied. Supplemental education appears to be more effective in large-volume laxatives (4 L PEG) used as bowel preparations, either in single or divided doses. A possible reason may be that high volume leads to reduced patient tolerance [53–55], while supplementary education could improve compliance. Regardless of year, country, diet, and assessment scale, supplemental education is positive in increasing rates of adequate bowel preparation in outpatients.

Consistent with improved rates of adequate bowel preparation, supplemental education also improved bowel quality scores (Fig 6). This is in line with a recent meta-analysis [56] that found that mobile health technology is associated with better bowel preparation quality scores. This is also in line with a meta-analysis published in 2021 [40], which showed that reinforced education increases colonoscopy BBPS scores and decreases OBPQS scores. Cecal intubation time and withdrawal time can be used as indirect indicators to measure the quality of bowel preparation. Our meta-analysis showed that supplementary education does not shorten the cecal intubation time, but it can shorten the withdrawal time. This is consistent with two previous high-quality randomized controlled trials [36,57]. The possible reason why supplementary education can shorten the withdrawal time but not the cecal intubation time is that the endoscopist carefully observes the intestinal tract when withdrawing [58]. Our meta-analysis shows that supplementary education does not improve the polyp detection rate of outpatient colonoscopy patients (38.1% vs 32.6%, p = 0.06). A previous meta-analysis also showed that educational videos cannot increase the detection rate of polyps [42]. Our meta-analysis also shows that supplementary education does not increase the detection rate of adenomas (32.0% vs 28.5%, p = 0.33). In fact, studies have pointed out that the quality of bowel preparation is not closely related to the detection rate of adenomas [59–61]. Our meta-analysis showed that supplementary education had no statistically significant difference in reducing the nonattendance rate of outpatient colonoscopy (9.5% vs 11.5%, RR:0.82, 95% CI: 0.72 to 0.94, $I^2$ = 38%, p = 0.10). A recently published meta-analysis also shows that mobile health technology cannot

reduce the no-show rate of colonoscopy [56]. It is worth noting that there was no obvious heterogeneity ($I^2$ = 38%, p = 0.09) in the research. If we refer to the previously published meta-analysis [62–64], we can choose the fixed-effect model, which will obtain the completely opposite result (S9 Fig). However, the PRISMA statement strongly discourages this approach [65]. Finally, our meta-analysis shows that supplementary education can increase the willingness to repeat outpatient care (91.9% vs 81.4%, p = 0.004).

The research has the following limitations: First, there was obvious heterogeneity in the research, and it was finally determined that the heterogeneity was caused by publication bias and bowel preparation regimen, which may affect the credibility of the results. Second, the research time span is long, and there are scale updates, which may affect the judgment of an adequate bowel preparation rate. Since it is impossible for the included studies to be double blinded, this may have a subjective influence on the results. Finally, a subgroup analysis showed that supplemental education cannot improve adequate colon preparation in some cases, which limits its widespread use.

## Conclusion

Supplementary education based on standard of care educational materials can significantly improve the quality of intestinal preparation for outpatients, shorten the withdrawal time and increase the willingness to repeat.

## Supporting information

**S1 Checklist. PRISMA 2020 checklist.**
(DOCX)

**S1 Fig. Summary of research risk assessment based on the Cochran risk assessment tool.**
(TIF)

**S2 Fig. Sensitivity analysis comparing the effects of supplementary education based on traditional education and traditional education alone on the adequate bowel preparation rate.**
(TIF)

**S3 Fig. Cumulative meta-analysis sorted by year of publication.**
(TIF)

**S4 Fig. Cumulative meta-analysis sorted by total sample size.**
(TIF)

**S5 Fig. Cumulative meta-analysis sorted by Jadad scale.**
(TIF)

**S6 Fig.** Sensitivity analysis comparing the effects of supplementary education combined with traditional education and traditional education alone on the adequate bowel preparation rate based on (A) publication year, (B) country, (C) bowel preparation regimen, (D) diet restriction, (E) supplementary education method, (F) quality evaluation scale and (G) Jadad score.
(TIF)

**S7 Fig. Funnel plot comparing the effects of supplementary education based on traditional education and traditional education alone on the polyp detection rate.**
(TIF)

**S8 Fig. Funnel plot comparing the effects of supplementary education based on traditional education and traditional education alone on the nonattendance rate.**
(TIF)

**S9 Fig. Forest plot comparing the effects of supplementary education combined with traditional education versus traditional education alone on the nonattendance rate based on a fixed effect model.**
(TIF)

**S1 File. PRISMA 2020 flow diagram for new systematic reviews which included searches of databases and registers only.**
(DOCX)

**S1 Table.**
(DOCX)

## Author Contributions

**Data curation:** Sixu Liu, Lijun Xiao, Xiaolan Liu, Kai Zhou.

**Formal analysis:** Shicheng Peng, Sixu Liu, Lijun Xiao, Xiaolan Liu, Muhan Lü.

**Investigation:** Shicheng Peng, Jiaming Lei.

**Methodology:** Shicheng Peng, Jiaming Lei, Wensen Ren.

**Project administration:** Jiaming Lei, Muhan Lü.

**Resources:** Wensen Ren.

**Software:** Lijun Xiao, Kai Zhou.

**Supervision:** Kai Zhou.

**Validation:** Shicheng Peng, Jiaming Lei.

**Visualization:** Sixu Liu.

**Writing – original draft:** Shicheng Peng, Sixu Liu.

**Writing – review & editing:** Shicheng Peng, Sixu Liu, Wensen Ren, Muhan Lü.

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
