## [Decision Letter · Decision Letter 0]

17 Jan 2022

PONE-D-21-37162Supplementary education can improve the rate of adequate bowel preparation in outpatients: a systematic review and meta-analysis based on randomized controlled trials.PLOS ONE

Dear Dr. Zhou,

Thank you for submitting your manuscript to PLOS ONE. After careful consideration, we feel that it has merit but does not fully meet PLOS ONE’s publication criteria as it currently stands. Therefore, we invite you to submit a revised version of the manuscript that addresses the points raised during the review process.

ACADEMIC EDITOR:I agree with the reviewers that the manuscript covers an interesting and important topic. However, I also agree with the reviewers that some issues need to be addressed.Apart from the reviewers´ concerns I would also recommend the following changes:- English should be edited by a native English speaker. I agree with reviewer 2 that verb tenses are not appropriate across the manuscript.- Page 4: reference 23 was published in 2017 (not in 2015)- Page 4: I would recommend to avoid journal names (Gastrointest Endosc)- Page 13: How many studies were analyzed? 15?, 21?- Can you explain the differences in cleansing quality between the different bowel solutions?

We look forward to receiving your revised manuscript.

Kind regards,

Antonio Z Gimeno-Garcia

Academic Editor

PLOS ONE

Journal Requirements:

(This study is independent research funded by the following grants: Special support (cultivation) for young scientific and technological talents of Southwest Medical University (No. 00031718) and Talent development project of The Affiliated Hospital of Southwest Medical University (N0. 20061).)

4. Please remove all personal information, ensure that the data shared are in accordance with participant consent, and re-upload a fully anonymized data set. 

Reviewers' comments:

Reviewer's Responses to Questions

**Comments to the Author**

1. Is the manuscript technically sound, and do the data support the conclusions?

Reviewer #1: Yes

Reviewer #2: Partly

2. Has the statistical analysis been performed appropriately and rigorously? 

Reviewer #1: Yes

Reviewer #2: I Don't Know

3. Have the authors made all data underlying the findings in their manuscript fully available?

Reviewer #1: Yes

Reviewer #2: Yes

4. Is the manuscript presented in an intelligible fashion and written in standard English?

Reviewer #1: Yes

Reviewer #2: No

5. Review Comments to the Author

Reviewer #1: I have read with great interest this exhaustive meta-analysis about the efficacy of different supplementary methods to increase adequate bowel preparation. The authors have analyzed 21 records with more than 11.000 patients with an evident increase in adequate bowel preparation (73% till 80%) and overall patients satisfaction.

As endoscopist my comments are related to obtain valid information to daily clinical practice:

1) As the authors have pointed they mixed records with high/low quality, different bowel preparation regimen and diet restriction. In my opinion supplementary education is crucial to increase efficacy, but probably the method is not the most important. Can the authors give more explanation about which is the best bowel cleaning protocol?

2) Regarding bowel preparation regiment, it is difficult to understand why Split 4L PEG-ELP or 2L PEG+ascorbate are inferior to Split 4L PEG or 4L PEG-ELP. Can you explain this it better?

3) Based on your results can you assuming low-fiber diet and low-residue diet are superior to clear liquid diet?

4) It is difficult to understand which is the best method?. I think we can not assume which is better because clinical protocols are very different. Can you elaborate more this is the discussion section?

Reviewer #2: I congratulate the authors for this excellent meta-analysis. However some issues limit its quality and need to be addressed.

Major comments

1. The language needs to be improved. The time tenses should be in the past, not in present tense as the analysis is already been done.

2. References should be improved. Ref 28, 35 and others are not properly addressed. Ref 16 and 33 refer to the same publication.

3. I would consider to analyze a ceiling effect. A ceiling effect in bowel cleansing (BC) is present. In an unselected population, it is very difficult to improve adequate BC higher than a given rate (90%-95%). That is, in general low rates of bowel cleansing may be improved better than high rates. The benefit of an educational measure intervention is probably greater when bowel cleansing is poor in controls. I table 2, you may notice that the benefit of the intervention is greater in studies with lower BC rates in the control group. I hypothesize that the benefit of any intervention (including educational measures) may be higher when the BC rates of the control group are low (<70%). I suggest to explore the benefit in two groups depending on the Bowel cleansing rates of the control population.

4. The publication bias effect should be further explained. Is it possible to identify which studies are more affected?

5. I don’t see the point in separating 4 liter PEG and 4 liter PEG-ELP. As far as I know PEG is not given without electrolytes. I also don’t see the point in separating low-residue diet and low-fiber diet. Low-fiber diet is a more accurate term, as the fiber is the only component of the diet that may be modified, but both terms refer to the same diet restriction. I will recommend to group PEG and PEG-ELP and to group low-residue and low-fiber diet.

6. The arguments pointed in the first paragraph of the discussion (lines 387-395) seem the justification and aims of this study. It seems that the first paragraph of the discussion is more suited in the introduction section.

Minor comments

1. Table 2. To facilitate the reading, I suggest to include de proportion in percentage, besides the raw numbers in the outcomes.

2. BP has usually lower quality between inpatients and outpatients. In fact, inpatient has been identified as a risk factor for poor BP in several studies. I don’t see the point of the comparison between inpatients and outpatients in several sentences. There is no also, no definite evidence that inpatients get, in general, better medical education than outpatients.

3. Line 48. “Supplementary education for outpatients based on standard of care (…) can significantly improve the quality of intestinal preparation”

4. Line 59. “As we all know” Is unnecessary and superfluous.

5. Major factors affecting the quality of BP are some diseases such as diabetes mellitus, chronic constipation and abdominal surgery; some drugs such as tryciclic antidepressants and opioids and previous episode of inadequate BP. They should be stated in the introduction when naming factors affecting the quality of BP.

6. Please cite and comment and updated recent meta analysis on reinforced education for bowel preparation.

Guo X, Li X, Wang Z, Zhai J, Liu Q, Ding K, Pan Y. Reinforced education improves the quality of bowel preparation for colonoscopy: An updated meta-analysis of randomized controlled trials. PLoS One. 2020 Apr 28;15(4):e0231888. doi: 10.1371/journal.pone.0231888. PMID: 32343708; PMCID: PMC7188205.

7. Line 150.The expression “And so on” imply that other groups are done, but are not describe. I suggest to eliminate “so on” or describe the how the grouping were performed.

6. PLOS authors have the option to publish the peer review history of their article (what does this mean?). If published, this will include your full peer review and any attached files.

Reviewer #1: No

Reviewer #2: **Yes: **Marco Antonio Alvarez-Gonzalez

---

## [Author Response · Author response to Decision Letter 0]

14 Mar 2022

Dear Editors,

Thank you very much for considering the publication of our manuscript. The academic editor and reviewers have made great comments on our manuscript. After discussion with all the coauthors, we revised this manuscript seriously and carefully according to academic editor and reviewers’ opinions and highlighted the changes to our manuscript in red font in MS Word. The answers to each reviewer are as follows:

Response to the Academic Editor’s comments:

English should be edited by a native English speaker. I agree with reviewer 2 that verb tenses are not appropriate across the manuscript.

Answer to question: Thank you very much for this comment. We agree with this point completely, and we have changed the verb tense to past tense.

Page 4: reference 23 was published in 2017 (not in 2015)

Answer to question: Thank you very much for this comment. We have corrected the errors on page 4.

Page 4: I would recommend to avoid journal names (Gastrointest Endosc)

Answer to question: Thank you very much for this suggestion. We truly agree with this. Therefore, we changed the following sentences: ‘A meta-analysis published on gastrointestinal endoscopy in 2015 showed that these methods can improve the quality of colonoscopy bowel preparation’ on page 4 of the “Introduction” section to ‘A meta-analysis published in 2017 showed that these methods improved the quality of bowel preparation for colonoscopy.’

Page 13: How many studies were analyzed? 15?, 21?

Answer to question: Thank you very much for this comment. To make the results more readable, we changed the following sentences: ‘According to the Jadad scale, the fifteen studies included in the analysis were of high quality, and the remaining six were of low quality.’ on page 13 of the “Risk of bias in studies” section into ‘According to the Jadad scale, fifteen of the studies included in the analysis were of high quality, and the remaining six were of low quality.’

Can you explain the differences in cleansing quality between the different bowel solutions?

Answer to question: Thank you very much for this comment. We fully agree with Reviewer 2. All administrations of polyethylene glycol as a cathartic were also administered with electrolytes, so the results of our subgroup analysis based on this have been updated in Supplementary Figure 2C and explained in the Discussion section. Since the included analyses did not compare different bowel preparation regimens, it is difficult to interpret which is better.

Response to Reviewer 1’ comments:

1. As the authors have pointed out, they mixed records with high/low quality, different bowel preparation regimens and diet restrictions. In my opinion supplementary education is crucial to increase efficacy, but probably the method is not the most important. Can the authors give more explanation about which is the best bowel cleaning protocol?

Answer to question 1: Thank you for your comment. Since the included analyses did not compare different bowel preparation regimens, it is difficult to say which is the best. At the same time, bowel preparation protocols include diet restriction and bowel preparation regimens, and direct comparative studies are lacking to determine which protocol is the most effective. For which bowel preparation regimen is the best, you can refer to a meta-analysis published in 2006 [1]. Of course, I think the latest network meta-analysis is needed for validation. As you said, “In my opinion, supplementary education is crucial to increase efficacy”, and we proved your point.

2. Regarding the bowel preparation regimen, it is difficult to understand why split 4 L PEG-ELP or 2 L PEG+ascorbate is inferior to split 4 L PEG or 4 L PEG-ELP. Can you explain this it better?

Answer to question 2: Thank you for this comment. Considering that in clinical practice all polyethylene glycol as a laxative was administered with electrolytes, we combined the PEG-ELP group with PEG and combined it into the PEG group. Subgroup analyses (Supplementary Figure 2C) based on this showed that supplemental education was statistically significant in improving the rate of adequate bowel preparation when 4 L polyethylene glycol was used as a laxative, regardless of single dose or divided dose. In fact, we did not compare bowel preparation adequacy between different laxatives, and subgroup analyses were performed to explore sources of heterogeneity.

3. Based on your results can you assuming low-fiber diet and low-residue diet are superior to clear liquid diet?

Answer to question 3: Thank you very much for this comment. We strongly agree with commenter 2 that low-fiber diet and low-residue diet are the same diet, the difference being that low fiber is more accurate [2]. We combined the low-residue diet and the low-fiber diet into one subgroup analysis, and the results are updated in Supplementary Figure 6D. The results show that supplemental education increases the rate of adequate bowel preparation regardless of diet. In fact, the meta-analysis did not compare the effect of different diets on the quality of bowel preparation, and subgroup analyses were performed only to explore sources of heterogeneity.

4. It is difficult to understand which is the best method? I think we can not assume which is better because clinical protocols are very different. Can you elaborate more this is the discussion section?

Answer to question 4: Thank you for your suggestion. We have carefully listened to and studied your suggestion, and we believe that we have demonstrated that supplemental education has achieved the goal of improving the rate of adequate bowel preparation in outpatients, and it is not the direction of our analysis as to which regimen is best. Of course, we discussed the different effects of supplemental education under different bowel preparation regimens in the Discussion section.

Response to Reviewer 2’ comments:

Major comments

1. The language needs to be improved. The time tenses should be in the past, not in present tense as the analysis is already been done.

Answer to question 1: Thank you for your comments. We agree with this point. We have changed the verb tense to past tense. We asked a native English speaker to help us correct our manuscript. After his careful correction, we are sure that there are no grammatical and spelling errors in our manuscript, and our manuscript is now qualified for publication in this journal.

2. References should be improved. Ref 28, 35 and others are not properly addressed. Ref 16 and 33 refer to the same publication.

Answer to question 2: Thank you for this comment. We use Endnote X9 for bibliographic management, and it may be a software bug that caused reference insertion errors. The references have been carefully checked for accuracy.

3. I would consider to analyze a ceiling effect. A ceiling effect in bowel cleansing (BC) is present. In an unselected population, it is very difficult to improve adequate BC higher than a given rate (90%-95%). That is, in general, low rates of bowel cleansing may be improved better than high rates. The benefit of an educational measure intervention is probably greater when bowel cleansing is poor in controls. Table 2 shows that the benefit of the intervention is greater in studies with lower BC rates in the control group. I hypothesize that the benefit of any intervention (including educational measures) may be higher when the BC rates of the control group are low (<70%). I suggest to explore the benefit in two groups depending on the Bowel cleansing rates of the control population.

Answer to question 3: Thank you for the suggestions. We agree with this point, so we performed subgroup analysis based on the rates of adequate bowel preparation (<70%) in the control group. As you estimated, our analysis (Figure 3) showed a ceiling effect in bowel preparation. Supplemental education increased adequate bowel readiness by 10.47% (60.53% to 71.9%, p< 0.00001) in the low-ratio group but only 4.53% (82.67% to 87.20, p= 0.003) in the high-ratio group.

4. The publication bias effect should be further explained. Is it possible to identify which studies are more affected?

Answer to question 4: Thank you for this comment. We agree with this point. Therefore, we added the following sentences to the “Discussion” section: “Reasons for publication bias include the following: studies with positive or statistically significant results are more likely to be published than those with negative or insignificant results, authors are more likely to publish studies with positive results in English-language journals, and authors are selective about the results reported by the protocol hide”.

5. I don’t see the point in separating 4 liter PEG and 4 liter PEG-ELP. As far as I know PEG is not given without electrolytes. I also don’t see the point in separating low-residue diet and low-fiber diet. A low-fiber diet is a more accurate term, as fiber is the only component of the diet that may be modified, but both terms refer to the same diet restriction. I will recommend to group PEG and PEG-ELP and to group low-residue and low-fiber diet.

Answer to question 5: Thank you for the suggestion. We agree with this point. We eliminated the PEG-ELP and low-residue groups, redid the analysis and updated the results in Supplementary Figures 6C and D. Even more gratifying was that meta-regression analysis after improved grouping found a source of heterogeneity.

6. The arguments pointed out in the first paragraph of the discussion (lines 387-395) seem the justification and aims of this study. It seems that the first paragraph of the discussion is more suited in the introduction section.

Answer to question 6: Thank you for this suggestion. We fully agree with you and removed this section from the Discussion section.

Minor comments

1. Table 2. To facilitate the reading, I suggest to include de proportion in percentage, besides the raw numbers in the outcomes.

Answer to question 1: Thank you for this suggestion. We fully agree with the above and make changes in Table 2.

2. BP is usually lower quality between inpatients and outpatients. In fact, inpatients have been identified as a risk factor for poor BP in several studies. I don’t see the point of the comparison between inpatients and outpatients in several sentences. There is also no definite evidence that inpatients receive, in general, better medical education than outpatients.

Answer to question 2: Thank you for this comment. We have removed the above statement.

3. Line 48. “Supplementary education for outpatients based on standard of care (…) can significantly improve the quality of intestinal preparation”

Answer to question 3: Thank you for this suggestion. We changed the following sentences: ‘Supplementary education based on standard of care educational materials can significantly improve the quality of intestinal preparation for outpatients, shorten the withdrawal time and increase the willingness to repeat rate’ on line 48 of the “Abstract” section to ‘Supplementary education for outpatients based on standard of care can significantly improve the quality of bowel preparation.’

4. Line 59. “As we all know” is unnecessary and superfluous.

Answer to question 4: Thank you for this suggestion. We completely agree with you and have deleted the following sentence: ‘As we all know, compared with inpatients, outpatients have fewer opportunities to contact their doctors and get adequate guidance.’ from line 59.

5. Major factors affecting the quality of BP are some diseases, such as diabetes mellitus, chronic constipation and abdominal surgery; some drugs, such as tryciclic antidepressants and opioids; and previous episodes of inadequate BP. They should be stated in the introduction when naming factors affecting the quality of BP.

Answer to question 5: Thank you for this comment. We completely agree with you, and we changed the following sentence ‘Factors affecting the quality of intestinal preparation of patients include education level, economic level, family relationship, tolerance of laxatives, professional level of instructors and patient's comprehension and cooperative degree’ on the “Introduction” part into ‘Factors affecting the quality of intestinal preparation of patients include education level, gender, economic level, family relationship, tolerance of laxatives, professional level of instructors, patient's comprehension and cooperative degree, previous abdominal or colonic surgery, diabetes mellitus obesity, chronic constipation, drugs (opioids, antidepressants) and neurologic diseases.’

6. Please cite and comment and updated recent meta analysis on reinforced education for bowel preparation. Guo X, Li X, Wang Z, Zhai J, Liu Q, Ding K, Pan Y. Reinforced education improves the quality of bowel preparation for colonoscopy: An updated meta-analysis of randomized controlled trials. PLoS One. 2020 Apr 28;15(4):e0231888. doi: 10.1371/journal.pone.0231888. PMID: 32343708; PMCID: PMC7188205.

Answer to question 6: Thank you for this suggestion. We cited and analyzed the article in the Discussion section.

7. Line 150. The expression “And so on” implies that other groups are done but are not described. I suggest to eliminate “so on” or describe the how the grouping were performed.

Answer to question 7: Thank you for this suggestion. We fully agree with you and change the following sentence: ‘Taking into account the diversity of supplementary education, we try to classify the following according to the main characteristics: smartphone applications (whether it is social software such as WeChat’s official account push or targeted development applications), video (regardless of whether the video acquisition form is offline or Online), text messages (either serial or targeted), telephone call(to communicate with patients via telephone voice), booklet (booklets designed to increase patient understanding), and so on.’ on the “Data extraction” part into ‘Taking into account the diversity of supplementary education, we try to classify the following according to the main characteristics: smartphone applications (whether it is social software such as WeChat’s official account push or targeted development applications), video (regardless of whether the video acquisition form is offline or Online), text messages (either serial or targeted), telephone call(to communicate with patients via telephone voice) and booklet (booklets designed to increase patient understanding).’

References

[1] J. J Y. Tan and J. J. Tjandra. Which is the optimal bowel preparation for colonoscopy a meta-analysis. Colorectal Disease 2006 May;8(4): 247-58.doi: 10.1111/j.1463-1318. 2006. 00970.x.

[2] Eleese Cunningham. Are low-residue diets still applicable? J Acad Nutr Diet. 2012 Jun;112(6):960. doi: 10.1016/j.jand.2012.04.005.

In addition to correcting this manuscript according to the reviewers’ advice, we also checked the manuscript time and time again to avoid grammatical or spelling errors throughout the article. We hope our corrected manuscript can be published in your journal soon. That will be our great honor.

Thank you very much. Best wishes!

yours sincerely,

Kai Zhou

---

## [Editor Report · Decision Letter 1]

28 Mar 2022

Supplementary education can improve the rate of adequate bowel preparation in outpatients: a systematic review and meta-analysis based on randomized controlled trials.

PONE-D-21-37162R1

Dear Dr. Zhou,

We’re pleased to inform you that your manuscript has been judged scientifically suitable for publication and will be formally accepted for publication once it meets all outstanding technical requirements.

Kind regards,

Antonio Z Gimeno-Garcia

Academic Editor

PLOS ONE

---

## [Editor Report · Acceptance letter]

1 Apr 2022

PONE-D-21-37162R1 

Supplementary education can improve the rate of adequate bowel preparation in outpatients: a systematic review and meta-analysis based on randomized controlled trials. 

Dear Dr. zhou:

I'm pleased to inform you that your manuscript has been deemed suitable for publication in PLOS ONE. Congratulations! Your manuscript is now with our production department. 

Kind regards, 

on behalf of

Dr. Antonio Z Gimeno-Garcia 

Academic Editor

PLOS ONE